# Non-destructive DNA extraction for recovering mitochondrial genomes from museum grasshopper specimens

Hao Tang [ORCID]◑, Jiayi Zou◑, Keyao Zhang, Huateng Huang [ORCID]*

College of Life Science, Shaanxi Normal University, Xi'an, China

◑ These authors contributed equally to this work.
* huanghuateng@snnu.edu.cn

## Abstract

Museum collections of grasshoppers contain valuable genetic data for evolutionary, ecological, and systematic studies, but destructive sampling and complex molecular techniques often limit their use. This study presents a simple, non-destructive DNA extraction protocol designed for dried grasshopper specimens, effectively balancing DNA yield with morphological preservation. We tested this method on specimens aged 6–43 years, exploring various lysis conditions to optimize DNA recovery while maintaining specimen integrity. The protocol successfully extracted sufficient DNA to assemble mitochondrial genomes for most samples using cost-effective low-coverage shotgun sequencing. DNA from younger specimens produced the best results, while older samples (over 40 years) showed some challenges, though post-collection damage did not significantly affect mitochondrial sequence assembly. Additionally, we provide practical bioinformatics recommendations for processing short reads from dried specimens. Requiring minimal molecular expertise and relying on standard sequencing services, this accessible method is well-suited for widespread adoption. By unlocking genetic insights from museum collections, it offers new opportunities to deepen our understanding of grasshopper evolution and ecological dynamics.

## Introduction

Grasshoppers, a common name encompassing multiple families within the orthopteran suborder Caelifera, is a highly diverse group of insects. There are approximately 7,000 species of true grasshopper (Orthoptera: Acrididae) species worldwide with about 11,000 known species in the suborder [1]. They can be found from grasslands and forests to deserts and mountainous regions, thriving across an extensive range of environments [2]. As primary herbivores in ecosystems, they play an essential role in the nutrient cycle [3–5]. It is also a group of insects influenced human society since the dawn of civilization [6]: while some species act as agricultural pests

**Data availability statement:** The data are held or will be held in a public repository on NCBI (GenBank accession: PX395670, PX395671, PX395672, PX395673)(GenBank accession: PP662486). Because some of sequences are the first report of the whole taxon, we set a held for these sequence that will not be released to the public until Sep 16, 2026. However, according to Genbank's rules, it will be immediately released if the accession numbers appear in print. Self-produce codes are available on github: https://github.com/Linnaeus001/Self-sequence-processing-scripts/blob/main/Count_polyG.py.

**Funding:** Initials of the authors who received each award: H. H. T. Grant numbers awarded to each author: 32270493 The full name of each funder: National Natural Science Foundation of China URL of each funder website: https://www.nsfc.gov.cn/ Did the sponsors or funders play any role in the study design, data collection and analysis, decision to publish, or preparation of the manuscript?: NO.

**Competing interests:** The authors have declared that no competing interests exist.

inflicting significant damage, others serve as a sustainable protein source in various cultures, contributing to food security and local economies. However, despite grasshoppers being among the most recognizable and familiar insects, their diversity, evolutionary history and ecological dynamics remain poorly understood.

Museomics, an emerging field driven by recent advances in sequencing technology [7], offers significant benefits for evolutionary and ecological studies of grasshoppers. First, extracting genetic data from dried museum specimens circumvents some of the challenges associated with fieldwork. Unlike swarming locusts, many grasshopper species are endemic to restricted ranges, and climate-induced habitat loss further impedes fresh sampling. Incorporating museum samples can significantly enhance the taxonomic coverage of phylogenetic studies [8–11], which will help resolve the persistent systematic controversies in this group [2, 12,13]. Secondly, genetic information from type specimens (holotypes or paratypes) can help clarifying controversial species boundaries. Museomics can even uncover cryptic new species within museum collections, as demonstrated in other taxonomic groups [14]. Lastly, museum collections offer time-stamped genetic data that yield crucial insights into population dynamics, allowing researchers to trace how grasshopper populations have shifted in response to environmental pressures over decades [15] and to delineate their adaptive trajectories.

Early studies extracting DNA from dried grasshopper specimens employed strongly detergent-based extraction techniques, such as using DTAB-CTAB to bind DNA and remove impurities, followed by chloroform extraction [16,17]. However, these detergents can act as PCR inhibitors, leading to failure in subsequent amplification steps [18,19]. Additionally, in terms of specimen handling, only one of the first studies avoided significant specimen damage by perforating the exoskeleton with small pins [16], whereas subsequent research routinely ground an entire leg to maximize extraction efficiency [15,20–23]. Nevertheless, preserving intact external morphology remains essential for museum conservation. Especially for precious type specimens that are essential for taxonomic reference, irreversible morphological damage must be avoided.

In this study, a simple nondestructive DNA extraction protocol was applied to dried grasshopper specimens of varying ages. We evaluated and compared different lysis conditions with regard to DNA yield and morphological preservation. Our findings demonstrate that this method permits minimal-damage sampling while generating sufficient DNA for high-throughput sequencing. Moreover, we successfully obtained mitochondrial genome sequences via low-coverage shotgun sequencing for the majority of specimens and provide recommendations for the bioinformatic analysis of short reads derived from dried samples.

## Method

### DNA extraction and sequencing

This study used eight dried, pinned specimens collected over a period of 6–43 years (Table 1). The pinned specimens were provided by the Zoological & Botanical Museum of Shaanxi Normal University (SNNU), Shaanxi.

Table 1. Specimen list and information of DNA extraction.

| SpecimenID[1] | Geographic origin | Species | Preservation | Age[2] | Pretreatment to extraction | DNA extraction[3] Concentration (ng/µl) | A260/ A280 | A260/ A230 | Peak[4] (bp) |
|---|---|---|---|---|---|---|---|---|---|
| 17STF | Douala, Cameroon | *Stenocrobylus festivus* Karsch, 1891 | Dried | 6 | Yes | 269.9 | 1.84 | 2.29 | 1929 |
| 09SIQ | Shaanxi Prov., China | *Sinopodisma qinlingensis* Zheng, 1996 | | 14 | | 238.4 | 1.77 | 2.09 | 122 |
| 99PIT | Henan Prov., China | *Pielomastax tenuicerca* Xia & Liu, 1989 | | 24 | No | 46.6 | 1.72 | 1.61 | |
| | | | | | Yes | 141.7 | 1.8 | 2.14 | 101 |
| 89PEE | Sichuan Prov., China | *Pedopodisma emeiensis* (Yin, 1980) | | 34 | No | 61.7 | 1.6 | 1.41 | |
| | | | | | Yes | 170 | 1.75 | 2.16 | 114 |
| 86PEE[5] | Sichuan Prov., China | *Pedopodisma emeiensis* (Yin, 1980) | | 37 | No | 52.7 | 1.76 | 1.43 | |
| | | | | | Yes | 96.1 | 1.73 | 2.22 | 101 |
| 85SIR | Guangxi Prov., China | *Sinopodisma rostellocerca* You, 1980 | | 38 | Yes | 67.7 | 1.48 | 1.67 | 68 |
| 80PEE | Sichuan Prov., China | *Pedopodisma emeiensis* (Yin, 1980) | | 43 | | 149.9 | 1.72 | 1.7 | 82 |
| 80SIG | Guizhou Prov., China | *Sinopodisma guizhouensis* Zheng, 1981 | | 43 | | 35 | 1.57 | 1.03 | 62 |
| Neg. control | | | | | Yes | 0.1 | 2.51 | −0.03 | |

[1] SpecimenID is a combination of collecting data and abbreviation of species name (e.g., 2017 collected, *Stenocrobylus festivus*–17STF), [2] Number of years between specimen collection and extraction, [3] The final DNA extraction was eluted in 50ul Buffer TET and quantitatize by NanoDrop spectrophotometer (Thermo Fisher Scientific Inc., Waltham, MA, USA), [4] Fragment length distribution was quantitatized by Qsep100 fragments analyzer (Bioptic Inc., Jiangsu, CN). [5] Fail for generating any raw reads.

We primarily adhered to the protocol established by Korlević et al. [24], which is designed for the extraction of DNA from historical Diptera specimens. This protocol incorporates the MinElute PCR Purification Kit to retain shorter DNA fragments, along with a low-salt lysis buffer specifically formulated for fragile specimens [25]. For each grasshopper specimen, one hind leg was excised and used for DNA extraction. Given the brittle and fragile nature of the dried exoskeleton, we assessed two different lysis settings to evaluate their effectiveness in DNA extraction and preservation of morphological integrity.

In the first lysis settings, we aimed to minimize damage to the specimens; the hind leg was immersed in 200 µl of lysis buffer and then incubated overnight at 56°C on a heat block without agitation. The second setting was similar to the pre-treatment procedures from Korlević's protocol. In this case, the hind leg was rehydrated for 3 hours at 37°C by placing it alongside a piece of half-wet paper towel within a 1.5ml centrifuge tube. Subsequently, a sterile insect mount pin was employed to pierce the metafemur of the leg at the base (i.e., the end connecting with the trochanter). Following this, the leg was incubated overnight at the same temperature, and the tube was vortexed at 400 rpm during the incubation period. After lysis, the hind leg was removed and dehydrated in 100% ethanol for 30 minutes before being affixed back onto the pinned specimen using white glue, while the resulting lysates were utilized for the subsequent steps (see S1 Fig and Suppl 1 for detailed procedures). One negative control was added in this setting without soaking any tissue.

The concentrations of DNA extraction were measured using NanoDrop spectrophotometer (Thermo Fisher Scientific Inc., Waltham, MA, USA), and fragment lengths were analyzed with the QSEP100 fragment analyzer. Except for the negative control, all samples with pretreatment were deliver to downstream analyses. The standard library construction method for genomic DNA was employed. Briefly, genomic DNA sample was added as input material of each sample but eliminate the fragmentation step. Then DNA were end-polished, A-tailed, and ligated with the full-length adapter for Illumina sequencing, followed by further PCR amplification. Purification was employed via Agencourt SPRIselect (BeckmanCoulter, USA). Library Sequencing was employed at Novogene Co., Ltd. (Beijing) utilizing the Novaseq X Plus 150 PE system.

## Sequence analyses

Sequencing reads were initially filtered using FASTP [26] to remove adapters and low-quality regions, and the percentages of filtered reads and bases were recorded. Over fragmented DNA of might result in long poly-G repeats in the end of reads, so we also calculated the percentage of reads containing more than 15 bases of poly-G sequence using the SeqIO module in Biopython [27]. Exogenous DNA contamination was assessed using the PlusPFP database and added desert locust *Schistocerca gregaria* database within Kraken2 [28]. Using: kraken2 --threads 16 --quick --paired --db/home/path/database --report sample.kreport R_1.fq.gz R_2.fq.gz to run the classification and generate the report.

The pattern of DNA damage is more pronounced at the 3' and 5' ends of DNA fragments, particularly within the outermost ~5 base pairs [29]. Hence, we DamageProfiler [30] to calculate G>A substitution frequencies for the first 15 bases of the 3' ends and C>T substitution frequencies for the 5' ends.

The assembly of complete mitochondrial genomes from low-coverage next-generation sequencing (NGS) data has become a common practice, facilitated by the development of numerous rapid assembly methods [31,32]. However, we found that assembling whole mitogenomes from sequencing data of dried specimens presented specific challenges. To address these, four assembly pipelines were systematically tested and their outcomes compared. The first three pipelines—MITObim [33], MitoZ [34], and MEANGS [35] —are widely used tools designed as "one-click" assemblers of mitogenomes from NGS data. These programs were executed with their default settings unless otherwise specified. MITObim employs a seed-based iterative baiting and mapping strategy to assemble mapped reads into a final complete mitogenome. Assemblies were initiated using mitochondrial genomes from the same or closely related species obtained from NCBI (see Table 1 for reference species names and accession numbers), and the process was run for 30 iterations. MitoZ, a seed-free approach, exploits the high depth of mitochondrial genome for assembly. The "all" subcommand was used, and the skip-filter option was activated to process data filtered with FASTP [26]. MEANGS, a recently developed seed-free method, extends contigs derived from self-discovery seeds using a trie-search algorithm; its "deepin" module was applied. The fourth approach was a custom consensus-based pipeline. Reads were mapped to reference mitochondrial genomes using BWA-MEM [36]. The resulting BAM files were sorted with SAMtools [37] and imported into Geneious Prime 2024.0.5 (https://www.geneious.com) to generate consensus sequences.

The authenticity of assembled sequences from seed-free software were verified by BLASTn [38] searches against the NCBI nucleotide database with default settings, and the species corresponding to the highest-scoring hit for each query sequence was recorded. For all assemblies, annotation was conducted with MitoZ, and the regions corresponding to 13 protein-coding genes (PCGs) were extracted, and their sequence similarity to reference mitogenome PCGs was calculated. To further evaluate the potential impact of DNA damage on sequence accuracy, mitogenomes were assembled using reads subjected to additional trimming. As DNA damage primarily affects the termini of fragments, Cutadapt [39] was used to remove 5 bases adjacent to adapter sequences from each read. The processed reads were then employed in the assembly pipelines. The 13 protein-coding regions were compared between assemblies generated from trimmed and untrimmed reads using SNP-sites [40]. The number of sites differ between different assemblies, particularly those fitting G-to-A or C-to-T damage patterns, was tallied.

We also employed the EAGER (ver. 2.5.3) [41,42], an ancient DNA working pipeline to work through the raw sequencing data with default command. Briefly, the pipeline utilizes several programs to deal with sequencing data. For examples, it employ AdapterRemoval [43] to remove adapters. Alignment, quality control and sorting was employed here using BWA [44], FastQC (https://www.bioinformatics.babraham.ac.uk/projects/fastqc/) and SAMTools [37]. Molecular damage patterns were detected via DamageProfiler [30].

## Result

Pretreatment prior to lysis showed a trend toward improving DNA extraction efficiency (paired t-test, P = 0.06; Fig 1A). DNA yield exhibited a significant negative correlation with specimen age (P = 0.05), likely reflecting irreversible DNA degradation

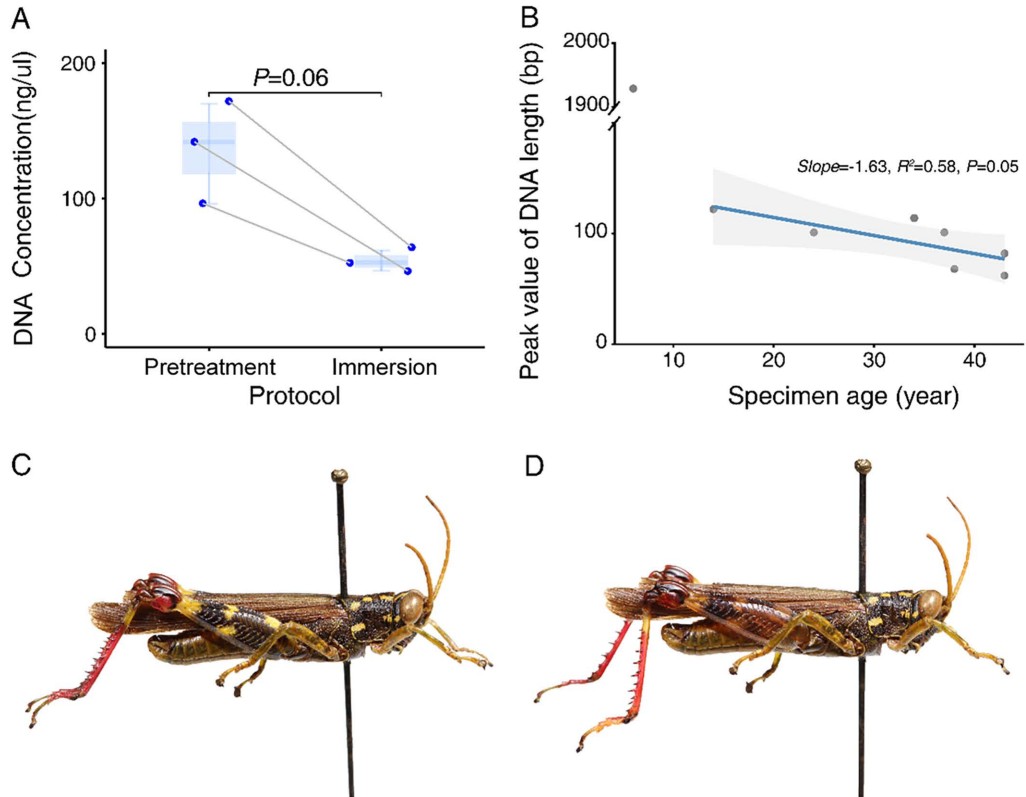

**Fig 1. DNA extraction result.** A. Extracted DNA concentration of two lysis settings from grasshopper specimens. P and E represents Pretreatment and non-pretreatment employment. Gray lines connect extractions from the two legs of the same specimen; B. The relationship between specimen age and the peak length of DNA fragments extracted. The blue line is the fitted linear regression, and the shaded area indicates the 95% confidence interval. C–D. Specimen 17STF before (C) and after extraction (D).

over time. Fragment analyzer results indicated that most DNA fragments were short, with a peak length of approximately 100 bp (S2 Fig). The sole exception was the youngest specimen, a 6-year-old *Stenocrobylus festivus* Karsch, 1891, which retained a notable proportion of fragments exceeding 1 kbp (S2 Fig). Older specimens generally yielded shorter fragments, as evidenced by a significant negative correlation between specimen age and peak fragment length, even after excluding the youngest sample (Fig 1B). The pretreatment caused minimal morphological alterations, limited to slight decolorization (Fig 1C, 1D). Trace DNA yield in negative control shows limited contamination introduction.

Standard Illumina library preparation and sequencing for genomic DNA was successful for all extractions except one specimen (Table 1). On average, nearly one-fourth of the reads (24.34%) were identified as exogenous DNA. However, substantial variation was observed among specimens. Two specimens had over half of their reads identified as contaminants, while the remainder exhibited contamination levels in the low teens. Notably, the proportion of exogenous DNA showed no significant correlation with specimen age (Fig 2B).

Due to the short length of the input DNA, a large number of reads contained adapter and polyG sequences. On average, approximately 20% of the reads included polyG sequences (defined as more than 15 consecutive Gs), and Fastp filtered out roughly 20% of reads and 30% of base pairs. Filtering outcomes differed slightly between the default setting (i.e., the --detect_adapter_for_pe option) and custom adapter sequences (Fig 3); the former failed to detect adapter sequences in several files, resulting in fewer base pairs being filtered (S1 Table). Significant variation was again observed

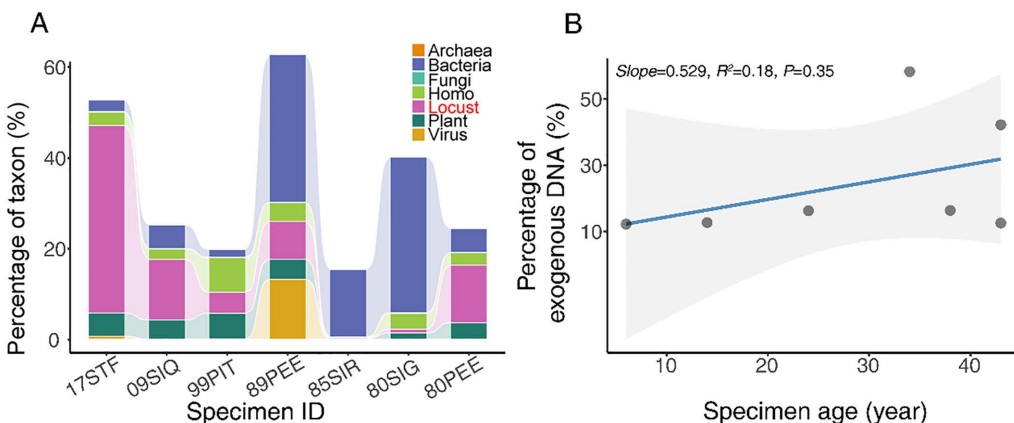

**Fig 2. Contamination of DNA.** A. Percentage of exogenous reads across specimens. Different colors represent different phylum. B. Relationship between specimen age and the percentage of exogenous DNA. The blue line is the fitted linear regression, and the shaded area indicates the 95% confidence interval.

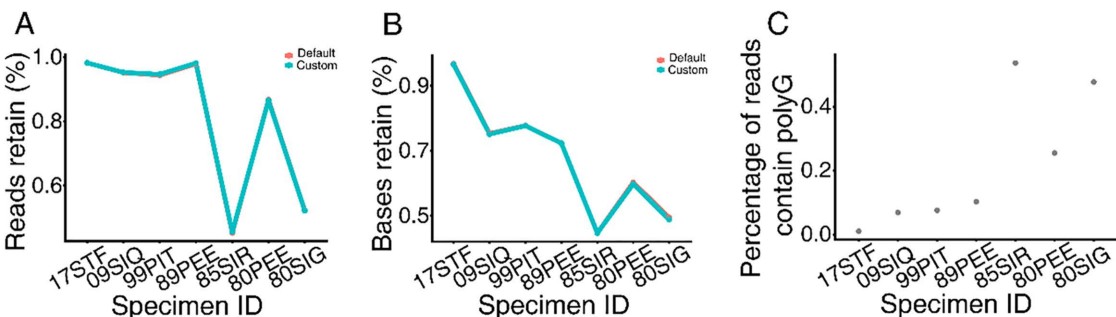

**Fig 3. DNA filtering result.** Percentage of reads (A) and bases (B) retained after filtering with FASTP. Red (cyan) line represents default (custom) setting. Percentage of reads contains polyG (C).

among specimens. Compared with the four younger specimens, three older specimens exhibited a marked increase in the percentage of polyG reads and the proportions of filtered reads and base pairs (Fig 3C).

Mitochondrial genome assemblies were easy for the four youngest specimens. We obtained the first mitochrondrial genome sequence for species *St. festivus* (GenBank accession: PP662486; S3 Fig). Among the methods tested, MitoZ exhibited the best overall performance— all of its assemblies have 13 complete protein-coding genes. MEANGS produced assemblies with more gaps in coding sequence (CDS) regions, while the reference-based MITOBim approach tended to generate longer sequences but failed for specimen 89PEE. Sequence similarity results were as expected: higher similarity when references are from the same or closely related species, and similarities were consistent across methods, confirming that all approaches assembled largely identical CDS regions. 09SIQ, 80PEE, 89PEE and 99PIT were also deposited (GenBank accession: PX395670, PX395671, PX395672, PX395673).

Assembly was more challenging for the three oldest specimens (80SIG, 80PEE, and 85SIR in S1 Table). One specimen, collected in 1985, which had the lowest number (353) of clean reads mappable to a reference sequence from the same genus, failed even with the mapping-consensus approach (S1 Table). MitoZ generated a few scaffolds for this specimen, but all were identified as bacterial mitogenome sequences through BLAST. For the two specimen collected in 1980, MitoZ and MEANGS partially recovered several CDS regions, while MITOBim was successful for only one of the two.

The EAGER pipeline produces series of result for each sample. All samples preform large proportion of full-length collapsed reads except for 80SIG and 85SIR who have large proportion of discarded reads indicating very low sequencing quality, and for 17STF, large proportion of uncollapsed paired reads represents a long insertion that exceeds the sequencing length (S4 Fig). 80SIG and 85SIR present low complexity contaminants due to a striking high level of duplication, corresponding to two of the failed mt-genome assembly (S4 Fig). Per base sequence content further reveals that all samples, except 17STF, show a distinct rise in G content (poly-G tails), reflecting 'no-call' bases in PE150 sequencing due to short insertions. In addition, 80SIG and 85SIR display uneven base composition at the beginning of reads, which likely represents overrepresented contaminant sequences (S4 Fig). No samples exhibit clear ancient damage pattern, C>T 5' and G>A 3' substitutions is relatively low (<20%) and without distinct trend, especially for 17STF, 80SIG and 85SIR (S5 Fig). To sum up, the EAGER results highlight that most libraries yielded reliable data suitable for downstream analysis, while 80SIG and 85SIR were strongly compromised by low sequencing quality, contamination, and duplication.

We also examined whether damage in dried specimen might bring in spurious substitutions onto the mitogenome assembly. However, conservatively trimming five base pairs from the beginning and the end of reads seems to cause little to no differences regarding the final assembled sequences, and most of the differences are not C-to-T or G-to-A changes (S2 Table).

## Discussion

Numerous studies have demonstrated the feasibility of retrieving genetic information from historical specimens, including those dating back to the 19th. For example, Lalonde et al. (2020) extract hDNA from mid- 19th century samples of *Junonia* butterfly specimen, and successfully recovered its mitochondrial haplotypes [45]. Zhang et al. (2020) extracted and assembled complete mitogenome from 140-year-old museum specimen of Malagasy butterfly *Malaza fastuosus* [46]. However, concerns about destructive sampling and the complexity of molecular workflows continue to dissuade curators from subjecting valuable specimens to genetic analyses. A simplified protocol that minimizes morphological damage while ensuring a high success rate could significantly advance the field of museomics.

This study demonstrates that a non-destructive method facilitates sufficient DNA recovery from museum grasshopper specimens while preserving external morphology. The pretreatment step is pivotal, as it restores specimen flexibility, thereby reducing femur breakage during puncturing and vortexing. Additionally, it minimizes debris generation during exoskeleton drilling, preventing silica spin column membrane blockage in subsequent steps. Unlike Dipteran specimens, which require elaborate reshaping (Korlević et al., 2021), grasshopper hind legs maintain structural integrity during lysis probably because their exoskeletons are more robust. Although the hind legs become "hollow" after extraction, they do not collapse and can be reattached to the specimen with minimal alteration to its external appearance (Fig 1D).

This protocol leverages standardized commercial kits, enhancing accessibility for researchers with limited molecular expertise. Silica spin column-based DNA extraction eliminates hazardous chemical handling, imposes minimal demands on pipetting precision, and reduces the total processing time to under two hours (excluding the initial incubation). This study employed the MinElute PCR Purification Kit rather than the commonly used DNeasy Blood & Tissue Kit due to its superior recovery of shorter DNA fragments (70 bp vs. 100 bp minimum, according to the manufacturer's manuals). Our fragment analyzer results validate this choice, as the peak DNA lengths of most dry specimens' approach or fall slightly below 100 bp (S2 Fig).

For next-generation sequencing (NGS), the initial DNA yield dictates subsequent methodological decisions. While advanced protocols for highly degraded DNA exist (e.g., single-stranded methods for ancient DNA [47]), their implementation typically requires more expertise in molecular lab work. Commercial kits designed for low-input DNA (<10 ng) are costlier and introduce PCR errors and clonality issues [48]. Maximizing the initial DNA yield is thus more cost-effective. Furthermore, prior research indicates that the DNA concentration extracted from historical specimens strongly correlates with the final amount of aligned data in target enrichment methods [49] and with the sequencing success of hyRAD libraries [22]. Our protocol yields 1,750–13,495 ng of DNA per specimen. This high yield not only supports the use of

mainstream library kits and commercial sequencing services but may also improve sequencing data quality. However, the relatively small and uneven sample size, together with the limited number of replicates across species, lysis buffers, and incubation times, reduces the statistical power and generalizability of our analyses, so results that are marginal or non-significant should be interpreted with caution and would benefit from validation with additional replicates in future studies.

Analyzing short reads derived from dried specimens presents greater challenges compared to fresh samples, particularly for collections older than 40 years. Although we did not conduct extensive sequencing across diverse specimen conditions, our analyses yielded several insights for future studies. First, a substantial proportion of reads may represent exogenous DNA, which should be considered when estimating required sequencing depth. Second, attention is needed during raw read preprocessing—specifically, using the -G option in FASTP to remove poly-G tails and verifying the adapter recognition. Third, DNA damage didn't occur rapidly post-collection, C>T and G>A substitutions frequent are relatively low at the 5' and 3' ends of the reads, respectively. Unusual fluctuate substitution on 17STF, 85SIR and 80SIG, probably due to remoted reference used and low complexity contamination (S5 Fig). There are mitigation strategies (e.g., USER-mix to cut out the uracil base or uracil-tolerant polymerase for PCR; [50,51]) for the DNA damage. We did not include such steps in this study, which makes the lab work simpler and cost less. The relieving result is that DNA damage does not seem to bring in extra artificial assembling errors for mitochondrial sequences (S2 Table). Sequencing nuclear genes, however, may require these additional mitigation strategies. Additionally, damage pattern also served as a tool to distinguishing Endogenous DNA from Contamination, because current contaminations showed a different deamination profile than the ancient sequences. Thus, although harsh preservation conditions bring severe fragmentation, making de novo assembly difficult, it confirms that the recovered sequences were authentic DNA we focus [52]. We also found that mitogenome assembly success varied across computational tools, with performance differing by specimen (S1 Table). Researchers could first use the mapping-consensus approach which can quickly exclude samples that do not have enough number of reads mappable to reference sequences, and then test multiple assembly programs. Additionally, tools such as CircularMapper can improve mapping on circular genomes by elongating the reference genome. However, in our dataset, using CM resulted in apparent mapping gaps in some samples with remoted reference. Therefore, we retained the default EAGER pipeline (BWA-aln) without specifying an alternative mapper. For consensus generation, although we choose PE150 sequencing strategy, trimmed reads are often much shorter than the nominal read length. In such cases, BWA-MEM may not always be the optimal aligner. While our comparisons did not reveal major improvement relative to the CM pipeline, we acknowledge that the effective read-length distribution after trimming is an important factor, and that short historical DNA fragments may sometimes be better accommodated by algorithms designed for degraded templates (e.g., BWA-aln or CM's internal mapper). This caveat should be considered when selecting mapping strategies for future historical DNA work.

In conclusion, this study demonstrates that non-destructive DNA extraction and sequencing from museum grasshoppers is achievable through an easy protocol. Unlocking genetic data from the extensive grasshopper collections in museums worldwide will substantially enhance our understanding of the evolutionary and ecological dynamics of this diverse clade. Future research incorporating larger sample sizes remains essential to further refine laboratory protocols and optimize downstream bioinformatic pipelines.

## Supporting information

**S1 Fig. Diagrams of DNA extraction workflow, showing the difference between lysis with the pretreatment (A) and minimal-disturbance setting (B).**
(TIF)

**S2 Fig. Fragment sizes of extracted DNA from dried grasshopper specimens.** The DNA of different specimen was distinguished by color. LM and UM represent lower (20 bp) and upper marker (1 kbp), respectively.
(TIF)

**S3 Fig. The assembled mitochondrial genome of *St. festivus*.** Cyan, red and orange represent protein-coding, tRNA and rRNA genes, respectively. Inner purple circle represents assembly depth.
(TIF)

**S4 Fig. Partial result of EAGER pipeline.** Discarded Reads, Sequence Duplication levels and per base sequence content calculated from raw reads before filtering.
(TIF)

**S5 Fig. DNA damage pattern of each sample from 3' and 5' end.**
(TIF)

**S6 Fig. Relationship between sample age and others exogenous DNA.** Linear regression lines, 95% confidence interval (shaded areas) and associated statistics are displayed.
(TIF)

**S1 Table. mtDNA genome assembly result using different assembler.** [1] Rows noted by "#" are specimen for which the adaptor sequence could not be detected on read2 by Fastp (v0.24.0). [2] Number of raw reads before filtering. [3] Reference sequeces used in MitoBim and the mapping-consensus approach. ***/**/* notes that the sequences are of the same species/genus/family as the specimen. [4] The top blast results of these 8 scaffolds are: *Timema bartmani* (83.3%), *Campylobacter jejuni* (99.9%), *Ignavibacterium album* JCM 16511 (85.71%), *Papio anubis* (73.91%), *Sus scrofa* (77.45%), *Naegleria fowleri* (86.54%), *Danio rerio* (73.33%), and *Caudoviricetes* sp. (92.5%). [5] Two specimens' assembled scaffolds ($) were subjected to BLAST search (blastn) on NCBI, but no significant matches were returned.
(XLSX)

**S2 Table. Sites difference between assembled sequence (only CDS) from trimmed and non-trimmed data.** [1] Sites match to "C/G (sequence from trimmed data) to T/A (sequence from non-trimmed data)" were regarded as damaged sites.
(XLSX)

**S1 File. Protocol for DNA extraction.** The full protocol is deposited at protocol.io (DOI: dx.doi.org/10.17504/protocols.io.bp2l6zj95gqe/v1).
(DOCX)

## Acknowledgments

We thank for Jire Xi (Shaanxi Normal University) for proposing experimental suggestions.

## Author contributions

**Data curation:** Hao Tang.

**Formal analysis:** Hao Tang, Jiayi Zou, Keyao Zhang.

**Funding acquisition:** Huateng Huang.

**Methodology:** Hao Tang, Jiayi Zou.

**Project administration:** Huateng Huang.

**Resources:** Huateng Huang.

**Software:** Hao Tang.

**Supervision:** Huateng Huang.

**Writing – original draft:** Keyao Zhang.

**Writing – review & editing:** Huateng Huang.

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
