## [Decision Letter · Decision Letter 0]

24 Jul 2025

Dear Dr. Huang,

We look forward to receiving your revised manuscript.

Kind regards,

Marta Maria Ciucani, PhD

Academic Editor

PLOS ONE

Journal Requirements:

[Initials of the authors who received each award�H. H. T.

Grant numbers awarded to each author: 32270493

The full name of each funder: National Natural Science Foundation of China

URL of each funder website: https://www.nsfc.gov.cn/

Did the sponsors or funders play any role in the study design, data collection and analysis, decision to publish, or preparation of the manuscript?: NO].

We note that one or more of the authors is affiliated with the funding organization, indicating the funder may have had some role in the design, data collection, analysis or preparation of your manuscript for publication; in other words, the funder played an indirect role through the participation of the co-authors. If the funding organization did not play a role in the study design, data collection and analysis, decision to publish, or preparation of the manuscript and only provided financial support in the form of authors' salaries and/or research materials, please do the following:

1. Review your statements relating to the author contributions, and ensure you have specifically and accurately indicated the role(s) that these authors had in your study. These amendments should be made in the online form.

2. Confirm in your cover letter that you agree with the following statement, and we will change the online submission form on your behalf:

“The funder provided support in the form of salaries for authors [insert relevant initials], but did not have any additional role in the study design, data collection and analysis, decision to publish, or preparation of the manuscript. The specific roles of these authors are articulated in the ‘author contributions’ section.

5. We note that Figures 1 and S3 in your submission contain copyrighted images. All PLOS content is published under the Creative Commons Attribution License (CC BY 4.0), which means that the manuscript, images, and Supporting Information files will be freely available online, and any third party is permitted to access, download, copy, distribute, and use these materials in any way, even commercially, with proper attribution. For more information, see our copyright guidelines: http://journals.plos.org/plosone/s/licenses-and-copyright.

1. You may seek permission from the original copyright holder of Figures 1 and S3 to publish the content specifically under the CC BY 4.0 license.

6. We notice that your supplementary figures are uploaded with the file type 'Figure'. Please amend the file type to 'Supporting Information'. Please ensure that each Supporting Information file has a legend listed in the manuscript after the references list.

7. Please include captions for your Supporting Information files at the end of your manuscript, and update any in-text citations to match accordingly. Please see our Supporting Information guidelines for more information: http://journals.plos.org/plosone/s/supporting-information

Reviewers' comments:

Reviewer's Responses to Questions

**Comments to the Author**

1. Is the manuscript technically sound, and do the data support the conclusions?

Reviewer #1: Yes

Reviewer #2: Partly

2. Has the statistical analysis been performed appropriately and rigorously?

Reviewer #1: Yes

Reviewer #2: No

3. Have the authors made all data underlying the findings in their manuscript fully available?

Reviewer #1: No

Reviewer #2: Yes

4. Is the manuscript presented in an intelligible fashion and written in standard English?

Reviewer #1: Yes

Reviewer #2: Yes

Reviewer #1: The paper by Tang et al presents a well-designed non-destructive DNA extraction protocol for museum grasshopper specimens. While the methodological advancements are clear, two major limitations should be addressed to strengthen the paper’s contribution.

First, the study lacks a rigorous assessment of endogenous DNA content. Although exogenous contamination is quantified using Kraken2, the proportion of authentic grasshopper DNA in the extracts remains unverified. Given the high variability in contamination levels (50% in some specimens), this raises concerns about the protocol’s efficiency in retrieving target DNA. Incorporating negative controls during extraction and quantifying endogenous DNA would provide critical validation.

Second, the title suggests a focus on mitochondrial genomes, yet the paper does not leverage these data for evolutionary or ecological insights. While the assembly of Stenocrobylus festivus mitogenomes is a valuable proof of concept, the opportunity to explore phylogenetic relationships, temporal genetic shifts, or comparative analyses with modern specimens is missed. To align the manuscript’s scope with its title, the authors should either reframe the title to emphasize methodology or include preliminary evolutionary analyses, such as a phylogeny or selection tests, to justify the current title.

Reviewer #2: The manuscript titled "Mitochondrial Genomes from Museum Grasshopper Specimens via Non-Destructive DNA Extraction," which addresses a timely and relevant topic in the field of museum genomics and minimally destructive DNA extraction from historical specimens. It presents a potentially valuable approach for recovering mitogenomes from museum specimens, however, in its current form, I do not consider the manuscript suitable for publication. The study contains several methodological and interpretative issues that undermine the scientific robustness and clarity of the work. I suggest revising these points about the scientific advancements:

1) Since the manuscript is intended as a methodological paper, the experimental procedures, including design, sampling, library preparation, and data analysis, should be described in much greater detail to allow proper assessment and reproducibility. As it stands, the methodological sections, in particular library preparation or data analysis, are too sparse to support a clear methodological advancement.

2) I also suggest moving the table S1 to the main text.

3) The diagram of the DNA extraction workflow (Figure S1) is well designed and highly informative. However, I recommend that the authors ensure full consistency between Figure S1, the main text, and Supplementary Table S1 with respect to the naming or labeling of the extraction methods applied to each sample. Currently, there appear to be some ambiguities in how the methodologies are referred to across different parts of the manuscript and supplementary materials, which could make it difficult for readers to follow or reproduce the experimental design.

4) This sentence can be better explained or reformulated: l. 84 “Due to the size constraints of a 1.5 ml centrifuge tube, the hind leg of each grasshopper specimen was initially removed”. It is not clear if the leg was totally removed from the tube, or when it was placed back in the tube.

5) Statistical support: some of the reported improvements are not statistically significant (e.g., P = 0.06), and yet are described in the text as if they were. Moreover, the statistical analyses employed should be more clearly described, and the authors should avoid overinterpreting marginal or non-significant results.

6) Inappropriate or incomplete analyses: the bioinformatic approach appears suboptimal or not fully adapted to the nature of the degraded DNA typically recovered from museum specimens and probably also to the chemistry of the sequencing platform. A useful pipeline could be represented by nf-core/eager (Fellows Yates et al., 2021). More justification for the chosen tools and pipelines, as well as a discussion of potential limitations, is needed. For example the authors report the presence of polyG-rich reads or sequences. However, it is unclear whether these observations reflect technical artifacts arising from either pipeline limitations or the sequencing chemistry due to 2-channel chemistry or read length (with PE150 sequencer may read into the adapter or generate low-quality tail regions, where polyG miscalls are frequent — especially in the absence of signal toward the end of the read).

7) More about bioinformatic approach: a method to improve mappings on circular genomes is CircularMapper. Moreover, more details about the parameters selected for reconstructing the consensus sequences need to be provided.

8) Low sample size and lack of controls: the most critical limitation of the study is the relatively small and uneven sample size, especially in relation to the number of species, lysis buffers, and incubation times tested. The low number of replicates per condition significantly limits the statistical power of the analyses and the generalizability of the findings. I strongly encourage the authors to explicitly acknowledge this limitation in the manuscript and to refrain from overinterpreting marginal or non-significant results. If possible, the inclusion of additional replicates would considerably strengthen the study’s conclusions.

9) Lack of negative controls (e.g., extraction blanks): they are not mentioned in the text. They usually can assess potential contamination when working with historical material.

10) In the discussion the authors stated that “Our protocol yields 1,750–13,495 ng of DNA per specimen, surpassing values reported for grinding-based protocols (e.g., 18.31–325 ng)”. I would strongly recommend removing or rephrasing this comparison, as it lacks sufficient contextual basis. The cited values come from a different study involving samples of unclear geographic origin, age, storage conditions, preservation history, or library preparation method, which may differ substantially from those used in the present study. If the authors wish to reference previously reported yields, it would be more appropriate to do so in a descriptive or contextual manner, without implying a direct performance comparison.

11) It appears that only one mitochondrial genome has been deposited in GenBank. Could the authors clarify why the remaining mitogenomes were not submitted, and whether there are plans to make them publicly available?

**Do you want your identity to be public for this peer review?** For information about this choice, including consent withdrawal, please see our Privacy Policy

Reviewer #1: No

Reviewer #2: No

---

## [Author Response · Author response to Decision Letter 1]

9 Oct 2025

Reviewer #1:

The study lacks a rigorous assessment of endogenous DNA content. Although exogenous contamination is quantified using Kraken2, the proportion of authentic grasshopper DNA in the extracts remains unverified. Given the high variability in contamination levels (50% in some specimens), this raises concerns about the protocol’s efficiency in retrieving target DNA. Incorporating negative controls during extraction and quantifying endogenous DNA would provide critical validation.

Response:

We thank the reviewer for this valuable suggestion. We now include the proportion of authentic grasshopper DNA in the Sankey plots (Fig. 2), providing a direct quantification of endogenous DNA content. Besides, it is important to note that variable contamination levels are a common feature in historical DNA studies, often reflecting specimen storage conditions rather than extraction performance. Despite some samples showing high exogenous fractions, our protocol consistently yielded sufficient endogenous DNA to assemble complete mitochondrial CDS. This demonstrates that the method is effective for recovering authentic target sequences, even under less-than-ideal preservation scenarios. We did add a negative control and quantified its dsDNA amount using Nanodrop, but didn’t sequence it. We add it in our table 1.

The title suggests a focus on mitochondrial genomes, yet the paper does not leverage these data for evolutionary or ecological insights. While the assembly of Stenocrobylus festivus mitogenomes is a valuable proof of concept, the opportunity to explore phylogenetic relationships, temporal genetic shifts, or comparative analyses with modern specimens is missed. To align the manuscript’s scope with its title, the authors should either reframe the title to emphasize methodology or include preliminary evolutionary analyses.

Response:

We agree with the reviewer’s assessment. To better reflect the methodological focus of the study, we have changed the title to:“Non-Destructive DNA Extraction for Recovering Mitochondrial Genomes from Museum Grasshopper Specimens.”This revised title avoids overstatement of evolutionary analyses while clearly emphasizing the protocol and its application.

Reviewer #2:

1) Since the manuscript is intended as a methodological paper, the experimental procedures, including design, sampling, library preparation, and data analysis, should be described in much greater detail to allow proper assessment and reproducibility. As it stands, the methodological sections, in particular library preparation or data analysis, are too sparse to support a clear methodological advancement.

Response:

We have expanded the methodological sections to provide greater detail on the library preparation procedures and contamination analyses. This now allows for more transparent assessment and reproducibility. Besides, a clear pipeline for DNA sampling is listed in supplemental file 1.

2) I also suggest moving the table S1 to the main text.

Response:

We have moved the former Supplementary Table S1 into the main text as Table 1.

3) The diagram of the DNA extraction workflow (Figure S1) is well designed and highly informative. However, I recommend that the authors ensure full consistency between Figure S1, the main text, and Supplementary Table S1 with respect to the naming or labeling of the extraction methods applied to each sample. Currently, there appear to be some ambiguities in how the methodologies are referred to across different parts of the manuscript and supplementary materials, which could make it difficult for readers to follow or reproduce the experimental design.

Response:

We carefully revised the manuscript to ensure consistent terminology across Figure S1, the main text, and the tables. We delete those unclear characters.

4) This sentence can be better explained or reformulated: l. 84 “Due to the size constraints of a 1.5 ml centrifuge tube, the hind leg of each grasshopper specimen was initially removed”. It is not clear if the leg was totally removed from the tube, or when it was placed back in the tube.

Response:

We revised the sentence to clarify the step.

5) Statistical support: some of the reported improvements are not statistically significant (e.g., P = 0.06), and yet are described in the text as if they were. Moreover, the statistical analyses employed should be more clearly described, and the authors should avoid overinterpreting marginal or non-significant results.

Response:

We deleted the term “significantly” where the P-value was marginal (e.g., P = 0.06) and revised the statistical descriptions to avoid overinterpretation.

6) Inappropriate or incomplete analyses: the bioinformatic approach appears suboptimal or not fully adapted to the nature of the degraded DNA typically recovered from museum specimens and probably also to the chemistry of the sequencing platform. A useful pipeline could be represented by nf-core/eager (Fellows Yates et al., 2021). More justification for the chosen tools and pipelines, as well as a discussion of potential limitations, is needed. For example, the authors report the presence of polyG-rich reads or sequences. However, it is unclear whether these observations reflect technical artifacts arising from either pipeline limitations or the sequencing chemistry due to 2-channel chemistry or read length (with PE150 sequencer may read into the adapter or generate lowquality tail regions, where polyG miscalls are frequent — especially in the absence of signal toward the end of the read).

Response:

We have incorporated the nf-core/eager pipeline to process our data, added corresponding method and results. We compare our previous DNA damage result, and revise its with eager’s output. For poly-G, we completely understand its relation to the length of DNA insertion, thus we change its expression in the Sequence analyses section.

7) More about bioinformatic approach: a method to improve mappings on circular genomes is CircularMapper. Moreover, more details about the parameters selected for reconstructing the consensus sequences need to be provided.

Response:

We tested the CircularMapper (CM) tool by installing it via Conda. Specifically, we applied CM to sample 99PIT and 17STF and compared the BAM results with our previous mapping approach. Here’s our main steps to generate the BAM via CM:

$java -jar /home/DataDisk/th_data/software/envs/CM/share/circularmapper-1.93.5-3/generator-1.93.5.jar -e 500 -i pi.zhengi.fasta -s "chrMT"

$bwa aln -t 4 pi.zhengi_500.fasta your.merged.fastqfile -n 0.04 -l 32 -f output.sai

$bwa samse pi.zhengi_500.fasta output.sai 99BTM_merge.fastq -f output_circmapper.sam

$java -jar /home/DataDisk/th_data/software/envs/CM/share/circularmapper-1.93.5-3/realign-1.93.5.jar -e 500 -i output_circmapper.sam -r pi.zhengi.fasta

We generate consensus sequence by Geneious prime, and map them to the reference. Although we found that in 99PIT, SNPs are fewer using CM, but appears a 196 bps gap in the sequence, and for 17STF, we find very few reads successfully mapped to the reference in 17STF using CM (We think it might cause by remoted relation between St. festivus and its reference Diabolocatantops_pinguis, for which is the only available reference of the same family). Although CM calls BWA-aln module, our sequencing strategy produce 150 bp pair end reads from historical DNA, and according to Dolenz’s research (Dolenz, 2024) BWA-mem still perform better toward 150 bp. So, we retained our original results, but further add a comment of CircularMapper.

8) Low sample size and lack of controls: the most critical limitation of the study is the relatively small and uneven sample size, especially in relation to the number of species, lysis buffers, and incubation times tested. The low number of replicates per condition significantly limits the statistical power of the analyses and the generalizability of the findings. I strongly encourage the authors to explicitly acknowledge this limitation in the manuscript and to refrain from overinterpreting marginal or non-significant results. If possible, the inclusion of additional replicates would considerably strengthen the study’s conclusions.

Response:

We agree and now explicitly acknowledge the limitation of sample size and replication in the revised Discussion.

9) Lack of negative controls (e.g., extraction blanks): they are not mentioned in the text. They usually can assess potential contamination when working with historical material.

10) In the discussion the authors stated that “Our protocol yields 1,750–13,495 ng of DNA per specimen, surpassing values reported for grinding-based protocols (e.g., 18.31–325 ng)”. I would strongly recommend removing or rephrasing this comparison, as it lacks sufficient contextual basis. The cited values come from a different study involving samples of unclear geographic origin, age, storage conditions, preservation history, or library preparation method, which may differ substantially from those used in the present study. If the authors wish to reference previously reported yields, it would be more appropriate to do so in a descriptive or contextual manner, without implying a direct performance comparison.

Response:

We did include a negative control. Although we did not sequence it, we quantified its dsDNA amount using Nanodrop. We now report this information in the revised Table 1 and discuss it in the result. Trace DNA yield in negative control shows limited contamination introduction.

11) It appears that only one mitochondrial genome has been deposited in GenBank. Could the authors clarify why the remaining mitogenomes were not submitted, and whether there are plans to make them publicly available?

Response:

No species of the genus Stenocrobylus had any sequences deposited in GenBank prior to our study; thus, we report it here for the first time. For the other species successfully assembled through consensus generation, corresponding sequences are already available in NCBI. We attempted to submit these as well, but received responses such as ‘Some or all of the sequences contain little similarity to other sequences in the database based on BLAST searches.’ Therefore, we now mark them as unverified, and they will be released to the public database upon publication.

---

## [Decision Letter · Decision Letter 1]

1 Dec 2025

Dear Dr. Huang,

Thank you for submitting your manuscript to PLOS ONE. After careful consideration, we feel that it has merit but does not fully meet PLOS ONE’s publication criteria as it currently stands. Therefore, we invite you to submit a revised version of the manuscript that addresses the points raised during the review process.

We look forward to receiving your revised manuscript.

Kind regards,

Marta Maria Ciucani, PhD

Academic Editor

PLOS ONE

Journal Requirements:

**Additional Editor Comments:**

Based on the reviewer's feedback, I am please to inform you that the editorial decision in Minor Revision.

The reviewer raised only a small number of comments that should be addressed to further improve the discussion and the manuscript.

Reviewers' comments:

Reviewer's Responses to Questions

**Comments to the Author**

Reviewer #2: All comments have been addressed

2. Is the manuscript technically sound, and do the data support the conclusions?

Reviewer #2: Partly

3. Has the statistical analysis been performed appropriately and rigorously?

Reviewer #2: Yes

4. Have the authors made all data underlying the findings in their manuscript fully available?

Reviewer #2: Yes

5. Is the manuscript presented in an intelligible fashion and written in standard English?

Reviewer #2: Yes

Reviewer #2: Dear Authors,

Thank you for your revisions. You have addressed all my previous comments, and I appreciate the improvements made to the manuscript. I only have a few additional suggestions that may further strengthen the work.

First, I recommend expanding the discussion of deamination patterns, also in light of the findings reported by Korlević et al. (2021) and Modi et al. (2021). Integrating these studies would provide useful context and help clarify how your results fit within the broader literature on ancient/historical DNA damage.

Second, I would like to comment on the statement:

“Although CM calls the BWA-aln module, our sequencing strategy produced 150 bp paired-end reads from historical DNA, and according to Dolenz’s research (Dolenz, 2024) BWA-MEM still performs better for 150 bp. So, we retained our original results, but further added a comment on CircularMapper.”

I do not fully agree with this reasoning. Your reads are indeed 150 bp as raw output from the sequencer; however, in practice they are very likely much shorter once adapter trimming is applied. Given the short insert sizes typical of historical DNA, the opposite adapter will often be reached again, producing artificially long reads with terminal poly-G stretches. After trimming, the reads used for mapping will therefore be substantially shorter than 150 bp. For this reason, BWA-MEM may not be the most appropriate choice in this context. I suggest clarifying this potential scenario in the manuscript and discussing its implications for the mapping strategy.

Overall, the manuscript is much improved, and I believe these additional clarifications would further enhance its accuracy and clarity.

**Do you want your identity to be public for this peer review?** For information about this choice, including consent withdrawal, please see our Privacy Policy

Reviewer #2: No

---

## [Author Response · Author response to Decision Letter 2]

23 Dec 2025

Reviewer #2:

First, I recommend expanding the discussion of deamination patterns, also in light of the findings reported by Korlević et al. (2021) and Modi et al. (2021). Integrating these studies would provide useful context and help clarify how your results fit within the broader literature on ancient/historical DNA damage.

Response:

We have expanded the Discussion to better describe DNA damage patterns observed in our study. We note that assessing these damage patterns helps distinguish endogenous DNA from contamination and informs mapping and assembly strategies. The manuscript has been revised accordingly.

Second, I would like to comment on the statement:

“Although CM calls the BWA-aln module, our sequencing strategy produced 150 bp paired-end reads from historical DNA, and according to Dolenz’s research (Dolenz, 2024) BWA-MEM still performs better for 150 bp. So, we retained our original results, but further added a comment on CircularMapper.”

I do not fully agree with this reasoning. Your reads are indeed 150 bp as raw output from the sequencer; however, in practice they are very likely much shorter once adapter trimming is applied. Given the short insert sizes typical of historical DNA, the opposite adapter will often be reached again, producing artificially long reads with terminal poly-G stretches. After trimming, the reads used for mapping will therefore be substantially shorter than 150 bp. For this reason, BWA-MEM may not be the most appropriate choice in this context. I suggest clarifying this potential scenario in the manuscript and discussing its implications for the mapping strategy.

Response:

We agree that, despite the nominal 150 bp read length, historical DNA libraries often produce substantially shorter trimmed reads because short inserts frequently reach the opposite adapter and generate poly-G tails. This scenario may indeed reduce the suitability of BWA-MEM, which is optimized for longer reads. In response to the reviewer’s suggestion, we have added a paragraph in the Discussion acknowledging this issue and clarifying that short trimmed reads may be better handled by algorithms tailored to degraded DNA, such as BWA-aln or the internal mapping module used by CircularMapper. We also note this as an important consideration for future applications of our pipeline. The manuscript has been revised accordingly. Thank you for all your proposal.

---

## [Editor Report · Decision Letter 2]

11 Jan 2026

Non-Destructive DNA Extraction for Recovering Mitochondrial Genomes from Museum Grasshopper Specimens

PONE-D-25-30043R2

Dear Dr. Huang,

We’re pleased to inform you that your manuscript has been judged scientifically suitable for publication and will be formally accepted for publication once it meets all outstanding technical requirements.

Kind regards,

Marta Maria Ciucani, PhD

Academic Editor

PLOS One
---

## [Editor Report · Acceptance letter]

PONE-D-25-30043R2

PLOS One

Dear Dr. Huang,

I'm pleased to inform you that your manuscript has been deemed suitable for publication in PLOS One. Congratulations! Your manuscript is now being handed over to our production team.

Kind regards,

on behalf of

Dr. Marta Maria Ciucani

Academic Editor

PLOS One